# Identifying a Suitable Model for Low-Flow Simulation in Watersheds of South-Central Chile: A Study Based on a Sensitivity Analysis

**Víctor Parra [1,2,\*], Jose Luis Arumí [1,2]**  **and Enrique Muñoz [3,4]**

1   Department of Water Resources, Universidad de Concepción, Chillán 3812120, Chile
2   Centro Fondap CRHIAM, Concepción 4070411, Chile
3   Department of Civil Engineering, Universidad Católica de la Santísima Concepción, Concepción 4090541, Chile
4   Centro de Investigación en Biodiversidad y Ambientes Sustentables CIBAS, Concepción 4090541, Chile
*   Correspondence: vmparra@ing.ucsc.cl; Tel.: +56-41-2345355

**Abstract:** Choosing a model that suitably represents the characteristics of a watershed to simulate low flows is crucial, especially in watersheds whose main source of baseflow generation depends on groundwater storage and release. The goal of this investigation is to study the performance and representativeness of storage-release process modeling, considering aspects such as the topography and geology of the modeled watershed through regional sensitivity analysis, in order to improve low-flow prediction. To this end, four groundwater storage-release structures in various watersheds with different geological (fractured and sedimentary rock) and topographic domains (steep and gentle slopes) were analyzed. The results suggest that the two-reservoir structure with three runoff responses is suitable (better) for simulating low flows in watersheds with fractured geological characteristics and rugged or steep topography. The results also indicate that a one-reservoir model can be adequate for predicting low flows in watersheds with a sedimentary influence or flat topography.

**Keywords:** modeling; sensitivity analysis; groundwater storage-discharge

## 1. Introduction

Hydrological models are a suitable tool for estimating water availability [1] and an effective tool for studying different hydrological processes at watershed scales such as precipitation, infiltration or groundwater storage-release. There are different types of hydrological models, including (i) conceptual and (ii) physical-based models. Conceptual models aim to reproduce processes through simple structures with parameters that conceptually represent a process (see Xu and Singh [2]; Chiew [3]). Meanwhile, physical-based models include the physical behavior of processes that occur in a watershed and use equations based on scientific principles based on known physical laws [4]. Physical-based models have the advantage of representing or modeling a watershed in a distributed manner, calculating the complete water balance of a watershed through physical equations and parameters that can be measured in the field. However, these models require a large quantity of information (data) that is often unavailable or difficult to measure [5]. Meanwhile, conceptual models have the advantage of requiring a lower quantity of input data (e.g., precipitation, temperature), but generate a simplified representation of the physical processes that occur in a watershed. Nonetheless, various recent investigations have shown that conceptual models can provide reliable models (e.g., Skaugen et al. [6]; Toledo et al. [7]; Muñoz et al. [8]; Parra et al. [9]); therefore, a conceptual model is a reliable alternative for studying hydrological processes in watersheds with limited hydrological information.

The study of hydrological processes is essential to understand the behavior of the basins. Among the hydrological processes, the groundwater storage-release process has been studied for quite some time due to its importance in baseflow generation in low-flow periods (see Wittenberg, [10]; Fenicia et al. [11]; Botter et al. [12]; Stoelzle et al. [13]). In conceptual hydrological models, the representation of the groundwater storage process is achieved using sub-models that consist of one or several reservoirs with a storage-discharge function (equation) [13], with the storage-discharge function in various models typically based on a linear relationship (e.g., T-M Model [14], abcd water balance model [15] or HYMOD model [16]). However, some authors mention that groundwater release is not a linear process, meaning that it must be represented as a non-linear process [10,17]. This entails a degree of uncertainty when choosing a hydrological model, as the groundwater storage structure or relationship (of the selected model) may not be suitable for simulating low-flow periods. Thus, there is a need to improve hydrological models for predicting low (or minimum) flows [18].

Recent studies have been done to improve low-flow prediction, for example, evaluating different hydrological models [18,19] or modifying the groundwater structure of the Soil Water Assessment Tools (SWAT model) [20,21]. Despite these contributions to improving low-flow prediction using hydrological models, authors such as Stoelzle et al. [13] mention a need to evaluate models for baseflow generation considering watersheds with different predominant geological characteristics. In fact, geological characteristics (e.g., fractures) determine the capacity of the riverbed to conduct water, allowing infiltration and groundwater storage [22]. Likewise, topographic characteristics (e.g., pronounced or less pronounced reliefs) are connected to groundwater movement [23] and influence it at various spatial scales [24]. Therefore, topography could have an important role in groundwater storage and release processes and low-flow generation, and it is fundamental that these characteristics (geological and topographic) be considered when choosing a storage-release structure when the purpose of the modeling is to simulate low flows. In addition, identifying the processes that take on greater importance in watersheds with different characteristics can contribute to the choice of a suitable model for simulating low flows. Thus, the objective of this investigation is to study-using a conceptual hydrological model the influence of geology and topography on groundwater storage-release process modeling through a regional sensitivity analysis in order to improve low-flow prediction in south-central Chile.

## 2. Methods

### 2.1. Study Area and Hydrometeorological Data

The study area comprised eight watersheds located in south-central Chile between ~36.5–37.0° and ~38.0–39.0° latitude South (Figure 1a). Watersheds without anthropogenic alterations or with minimal alterations were selected in order to avoid anthropogenic effects in the analysis. The selected watersheds in the central zone had a Mediterranean climate (~36.5–37.0°), while those in the south of the study area (~38.0–39.0°) had a wet climate [25]. The watersheds had a hydrological regime dominated by precipitation in winter and high precipitation variability, with annual averages from 700 to 3000 mm [8,25,26]. Watersheds without hydrological alteration or with minimum hydrological alteration (hydropower centers or reservoirs) were selected with the aim of studying the low-flow generation process.

Three watersheds, Chillan at Esperanza (CHE), Diguillín at San Lorenzo (DSL) and Cautín at Rari-Ruca (CR), were monitored on the western slope of the Andes and located at an elevation of around 700 m.a.s.l (Figure 1a). Most of the watersheds presented formations associated with volcanic and volcano-sedimentary sequences [27,28]. In addition, due to the tectonic uplift of the Andes [29], most of the watersheds presented rugged topography with steep slopes ranging from approximately 52.5° to 69.6° (Figure 1b,c). Three other watersheds, Chillan to Confluencia (CHC), Diguillín at Longitudinal (DL) and Cautín at Almagro (CA) were monitored downstream of CHE, DSL and CR, respectively, in the central depression of Chile (known as the Central Valley), and the monitoring stations were located

at about 80 m.a.s.l. Despite the dominance of formations of a volcanic origin in their upper parts, the middle and lower zones were characterized by the existence of sedimentary deposits (e.g., alluvial, colluvial and fluvioglacial deposits from the Andes Mountains [27]. In addition, the watersheds presented a topography with gentle slopes in their lower sections (Figure 1b,c). The Allipen at Los Laureles (ALL) watershed was monitored at an altitude of ~400 m.a.s.l. It presented a greater influence associated with volcanic sequences (55%) and a greater proportion of topography with steep slopes (Figure 1b,c). In contrast, the Quino at Longitudinal (QL) watershed was located in the Central Valley between ~1600 and 250 m.a.s.l. The watershed was formed by pyroclastic deposits (associated with volcanic sequences [27]) and presented a topography with gentle slopes (Figure 1b,c). Due to its altitudinal location, geologic formation (sedimentary volcanic deposits) and topography (gentle slopes), QL was classified as a sedimentary basin.

Table 1 shows the percentage of the geological formations and hydro-meteorological data in each watershed modeled. According to the predominant geology and relief present in each watershed, it was classified as volcanic-steep or sedimentary-gentle. Thus, CHE, DSL, CR and ALL were classified as volcanic-steep watersheds. CHC, DL, CA and QL were classified as sedimentary-gentle watersheds. Based on this classification were modeled the watersheds.

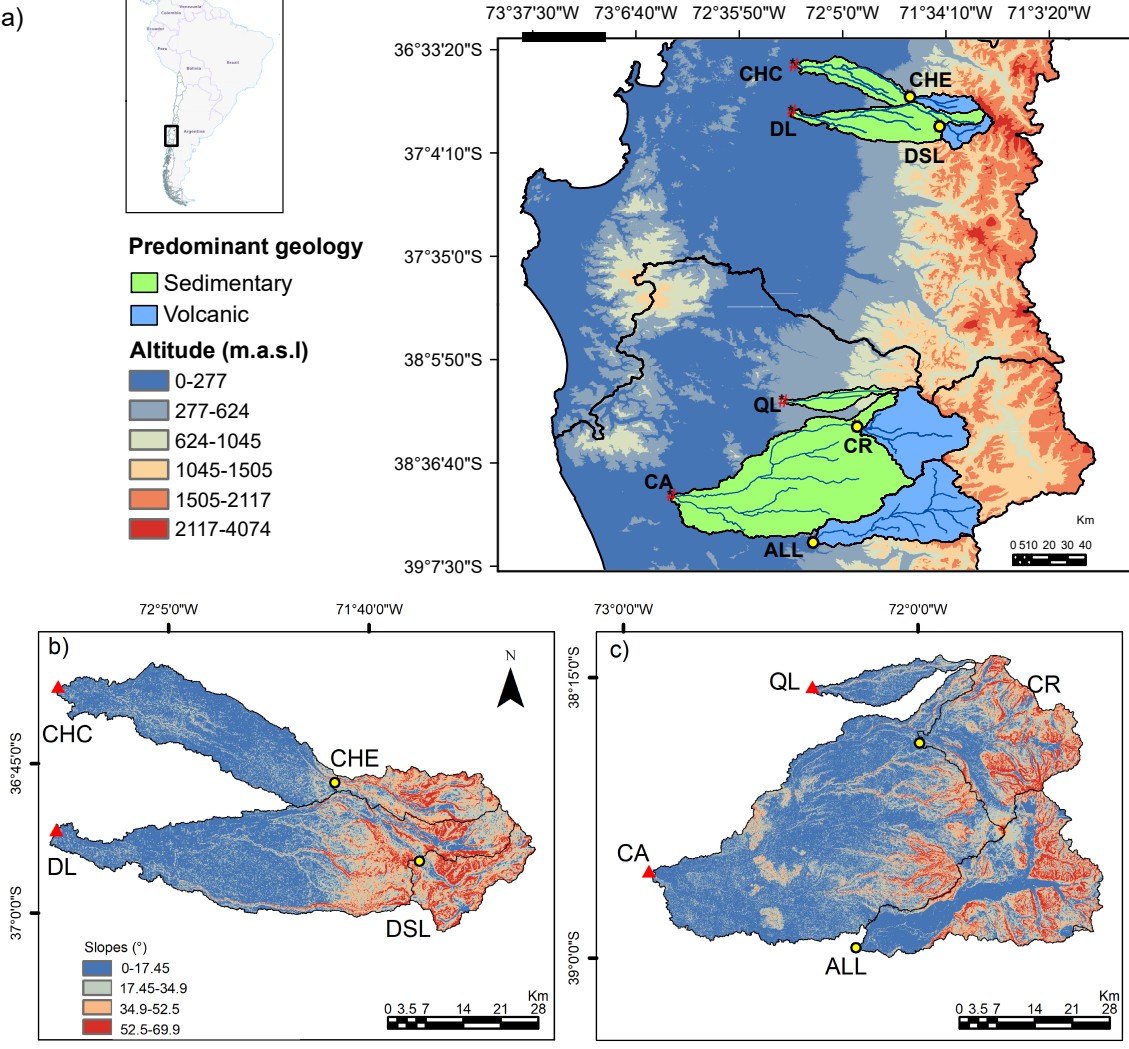

**Figure 1.** Study area and elevation map (**a**). Yellow circles correspond to volcanic watersheds and red triangles to sedimentary watersheds. Additionally, a slope map of watersheds (**b**,**c**) is shown.

**Table 1.** Monitoring stations, availability, geological formation and hydro-meteorological information in study watersheds.

| Station (ID) | | Area | Geological Formation (%) | | | Relief (°) | Hydro-Meteorological Information | | | |
|---|---|---|---|---|---|---|---|---|---|---|
| ID | Station | (km$^2$) | V | S | O | AS | MAP | MAD | MAT | MAE |
| CHE | Chillan at Esperanza | 210 | 90.4 | 0 | 9.6 | 17.8 | 2200 | 15.6 | 9.3 | 964 |
| DSL | Diguillín at San Lorenzo | 207 | 85.1 | 0 | 14.9 | 23.6 | 2300 | 16.4 | 9.2 | 920 |
| CR | Cautin at Rari-Ruca | 1255 | 96.6 | 1.2 | 2.2 | 14.4 | 2330 | 102.7 | 8.2 | 1006 |
| ALL | Río Allipen at Laureles | 1652 | 56.3 | 21.9 | 21.8 | 13 | 2294 | 139.4 | 8.7 | 1023 |
| QL | Quino at Longitudinal | 298 | 31 | 69 | 0 | 2.4 | 1850 | 13.1 | 12.5 | 1066 |
| CHC | Chillán to Confluencia | 754 | 25.3 | 70.4 | 4.3 | 8.1 | 1500 | 29.9 | 12.1 | 1163 |
| DL | Diguillin at Longitudinal | 1239 | 26.5 | 69.7 | 3.8 | 10 | 1736 | 46.9 | 10.9 | 1103 |
| CA | Cautin at Almagro | 5470 | 58.1 | 40.1 | 1.8 | 6 | 1838 | 261 | 10.5 | 1052 |

V: Volcanic; S: Sedimentary; O: Other; AS: Average slope; MAP: Mean annual precipitation (mm); MAD: Mean annual discharge (m$^3$/s); MAT: Mean annual temperature (°C); MAE: Mean annual evapotranspiration (mm). The statistics of hydro-meteorological data were obtained from the historical database of stations controlled by the DGA.

Modeling was carried out in order to analyze the representativeness of various storage-release sub-models implemented in the HBV model. Hydrometeorological (streamflow, precipitation and temperature) data on a daily time scale were required for the analysis. Due to data availability, a 10-year period of records was used for the analysis. The streamflow and precipitation records were obtained from the database managed by the General Water Directorate (DGA). In addition, as there were no continuous temperature series at nearby stations, (daily) series were obtained from the AgMERRA database [30]. AgMERRA is an open-access dataset with a resolution of 0.25° (~25 km). The Thornthwaite method [31] was used to estimate potential monthly evapotranspiration while daily evapotranspiration was calculated based on the HBV model method. The spatial distribution of precipitation, temperature and potential evapotranspiration throughout each watershed was obtained using the inverse weighted distance method (IDW) [32].

*2.2. HBV Hydrological Model Description*

To achieve the objective the HBV, the model was used and modified to analyze different groundwater storage-release sub-models. A sensitivity analysis was carried out and the performance of four groundwater storage and release sub-models or structures (including that used by default) in watersheds with different geological and topographic influences was evaluated.

The HBV model is a conceptual snow-rain water balance model. In this study, the simplified version of this model, developed by Aghakouchak and Habib [33] and based on Bergström [34], was used. The model simulates daily discharge based on daily precipitation, temperature and potential evapotranspiration time series [34] and includes a snow routine, a soil routine and a response routine (see conceptual diagram in Figure 2).

Precipitation was deemed as snow or rain depending on the temperature on the corresponding day above or below a threshold temperature (TT) equal to 0 °C. All precipitation was snow when the temperature was below TT, and was multiplied by a snow accumulation correction factor ($S_f$). All the snow contributed directly to snow storage (SS). If the actual temperature was greater than TT, there was snowmelt. Snowmelt water was controlled by a degree-day factor ($C_{melt}$), which determines the daily amount of melted snow depending on the difference between the actual and threshold temperatures. Subsequently, the sum of precipitation and snowmelt (ΔP) passed to the soil routine, which included two modules. The first module calculated the actual evapotranspiration ($E_a$), which

was equal to potential evapotranspiration (PET) if the relationship between soil moisture and maximum soil moisture (SM/FC) was above a threshold value for potential evapotranspiration (LP). On the other hand, for soil moisture values below LP, the actual evapotranspiration will be linearly reduced, as shown in Figure 2 (upper left corner).

The HBV model incorporates a routine to calculate daily evapotranspiration (PET) from monthly values. As inputs, the routine needs the long-term monthly mean potential evapotranspiration ($PET_m$) obtained from the Thornthwaite method, long-term monthly temperature averages ($T_m$) and daily mean air temperature (Td). The daily evapotranspiration was calculated by transforming (adjusting) the $PET_m$ through the difference between the $T_d$ and $T_m$ and a coefficient C (see Equation (1)). Bergström [34] mentioned that the adjusted potential evapotranspiration is limited to positive values and is not allowed to exceed twice the monthly average.

$$PET = (1 + C \cdot (T_d - T_m)) \cdot PET_m. \tag{1}$$

Subsequently, the model calculated runoff ($\Delta Q$), which depended on precipitation ($\Delta P$), the actual water content of the soil (SM), the maximum soil moisture (FC) and an empirical coefficient ($\beta$), which determined the relative contribution of rain or snowmelt to runoff (see upper left corner of Figure 2).

Finally, the runoff response routine estimated the runoff at the watershed outlet. The system consisted of two storage compartments, one above the other, which were directly connected to each other through a constant infiltration rate ($Q_{perc}$). The upper deposit had two outlets ($Q_0$ and $Q_1$), while the lower deposit had one ($Q_2$). When the water level in the upper deposit exceeded a threshold value (L), runoff was produced quickly in its upper part ($Q_0$). The response of the other outlets was relatively slow. The streamflows were controlled by recession coefficients $k_0$, $k_1$ and $k_2$, which represented the response functions of the upper and lower deposits. The constant infiltration rate ($Q_{perc}$) was controlled by a coefficient $k_p$.

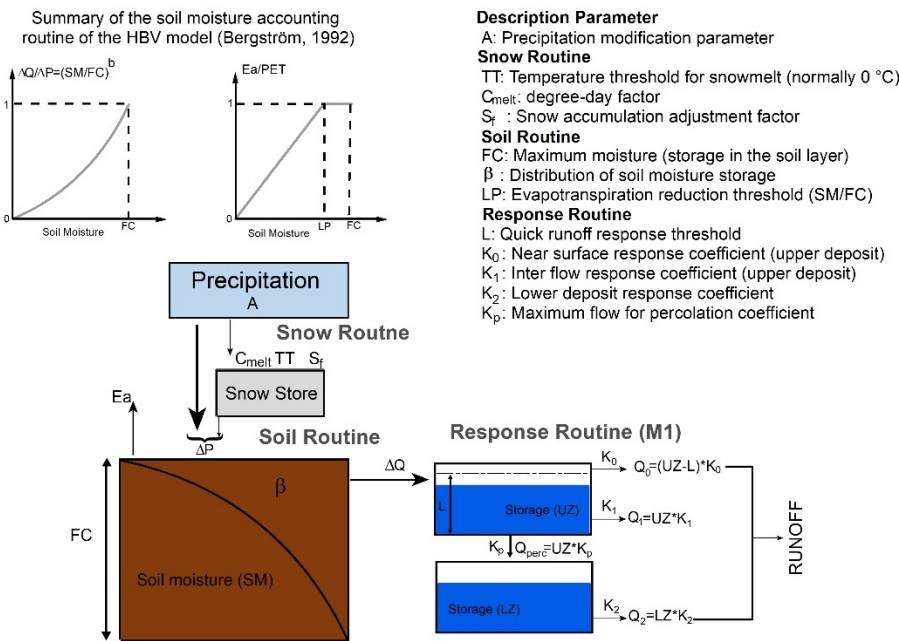

**Figure 2.** General conceptual diagram of the simplified HBV model, including a description of its parameters and main equations.

In order to ensure that the surface runoff process is quicker than the subsurface and groundwater runoff, the initial value of $k_0$ must always be greater than $k_1$. In addition, the response of the third outlet (groundwater runoff) ($Q_2$) must be slower than that of the second one ($Q_1$); therefore, $k_2$ must be lower than $k_1$ [33]. For a better understanding of the model, see Bergström [34] and Kollat et al. [35].

### 2.3. Analyzed Groundwater Storage Structures

Four groundwater storage and release configurations were analyzed. One structure was that used by default by the HBV model, while the other three were adapted based on conceptual models described in the literature (e.g., Wittenberg, [10]; Stoelzle et al. [13]). A description of each structure is provided below. Thus, the HBV model was adapted by changing the storage-release sub-models.

The first structure analyzed (M1) is that used by default in the HBV model. It was a two-reservoir structure described in the previous section (Figure 2).

The second structure (M2) consisted of one reservoir with two outlets ($Q_0$ and $Q_1$). Outlet $Q_0$ represented surface runoff that was produced if the water level of the reservoir surpassed a threshold value (L). Outlet $Q_1$ represented groundwater release from the aquifer with a linear relationship. Both outlets were controlled by recession coefficients $k_0$ and $k_1$ (Figure 3a).

The third structure (M3) represented the combination of two M2 structures in parallel (Figure 3b), which were connected by a parameter $\alpha$ (based on Reference [13]) that distributed recharge between them. Both reservoirs had two outlets, the upper two quick response and the lower two slow response. Parameter $\alpha$ was formulated in accord with Stoelzle et al. [13] and took values between 0 and 0.5. The smaller $\alpha$ was, the greater the recharge that entered the quick response reservoir (see diagram and formulas in Figure 3b).

The fourth structure (M4) was only one reservoir (Figure 3c). This model was similar to M2, but the groundwater storage-release equation (Q1) was based on a non-linear relationship (equation), as suggested by Wittenberg [10].

The performance of the HBV model under different groundwater storage structures (M1, M2, M3 and M4) was calculated for the calibration and validation periods. To analyze the performance of the models the $LOG_{NSE}$ function (described in the following section), which focused on the simulation of low flows, was used.

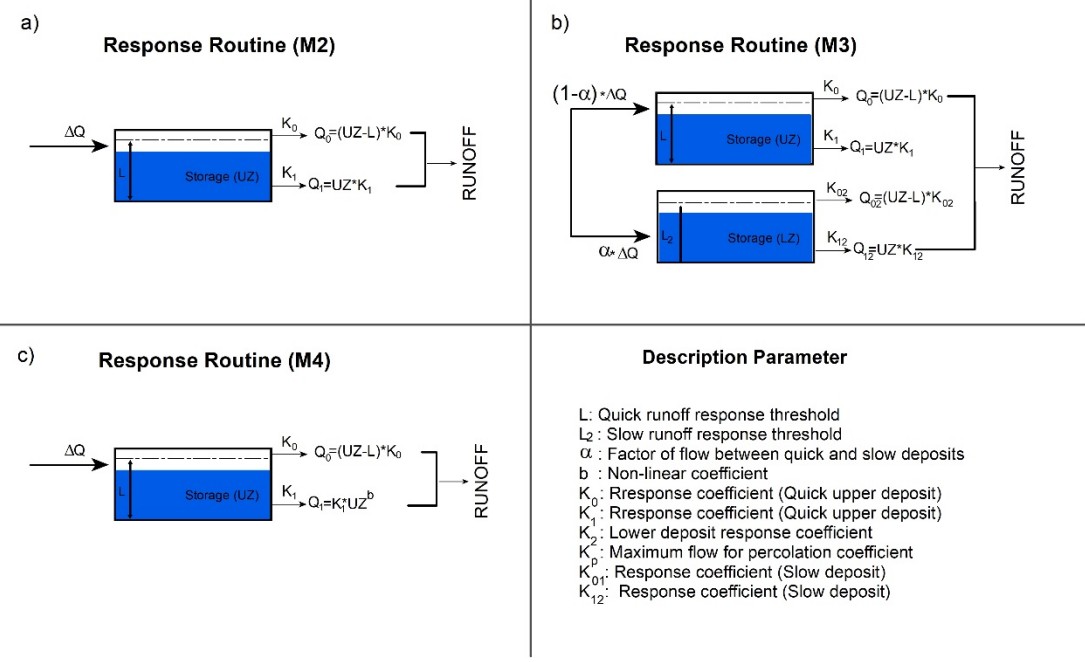

**Figure 3.** Additional adapted configurations of conceptual groundwater models, including a description of their parameters and main equations. (**a**) M2 model: one reservoir with two outlets; (**b**) M3 model: combination of two parallel reservoir with two outlets each one; (**c**) M4 model: one reservoir with two non-linear outlets.

Table [2] presents the calibration parameter ranges defined in accord with the conceptual representation of each and experience acquired in a prior study on Chilean watersheds [9]. Considering that at the Andes mountain range there was a low density of rain gauges because most were located in low-altitude areas, consequently, the spatial distribution of precipitation was not properly recorded [36]. Thus, a precipitation adjustment parameter (parameter A) was included to correct the underestimation of precipitation due to the orographic effect in the studied watersheds. This factor allows the model to obtain a long-term mass balance [8,37].

**Table 2.** Calibration parameter ranges for each model.

| Parameter (Units) | M1 | M2 | M3 | M4 |
|---|---|---|---|---|
| *Mass Balance* | | | | |
| A | 0.8–2.5 | 0.8–2.5 | 0.8–2.5 | 0.8–2.5 |
| *Snow Routine* | | | | |
| $C_{melt}$ $\left( mm\,°C^{-1}d^{-1} \right)$ | 0.5–7 | 0.5–7 | 0.5–7 | 0.5–7 |
| $S_f$ | 0.5–1.2 | 0.5–1.2 | 0.5–1.2 | 0.5–1.2 |
| *Soil Routine* | | | | |
| FC (mm) | 0–2000 | 1–2000 | 0–2000 | 1–2000 |
| $\beta$ | 0–7 | 0–7 | 0–7 | 0–7 |
| $a$ | - | - | 0–0.5 | - |
| LP | 0.3–1 | 0.3–1 | 0.3–1 | 0.3–1 |
| C ($°C^{-1}$) | 0.01–0.3 | 0.01–0.3 | 0.01–0.3 | 0.01–0.3 |
| *Response Routine* | | | | |
| L (mm) | 0–100 | 0–100 | 0–100 | 0–100 |
| L2 (mm) | - | - | 0–100 | - |
| $k_0$ ($d^{-1}$) | 0.3–0.6 | 0.3–0.6 | 0.3–0.6 | 0.3–0.6 |
| $k_1$ ($d^{-1}$) | 0.1–0.2 | 0.1–0.2 | 0.1–0.2 | 0.1–0.2 |
| $k_2$ ($d^{-1}$) | 0.01–0.1 | - | - | - |
| $k_p$ ($d^{-1}$) | 0.01–0.1 | - | - | - |
| $K_{02}$ ($d^{-1}$) | - | - | 0.3–0.1 | - |
| $K_{12}$ ($d^{-1}$) | - | - | 0.2–0.05 | - |
| b | - | - | - | 1–0.33 |

## 2.4. Sensitivity Analysis and Calibration

In order to understand the dependence of the parameters on the response of each model, a regional sensitivity analysis was carried out. The sensitivity analysis for everybody (SAFE) Matlab tool was used for the analysis [38].

Recent studies have shown that with a short time window (e.g., 5 years) good results are obtained from models in the calibration stage [39–42]. Considering these studies and data availability in each watershed, a calibration period of 6 years (April 2000–March 2006) was used for all the watersheds, except for Chillán at Esperanza, which had records up to 1994. Thus, for this watershed April 1980–March 1986 was used for the calibration. For the validation period, the four years after the calibration period (April 2006–March 2010 and April 1986–March 1990, respectively), was used. In both periods (calibration and validation), the first year of records was used to warm up the model, in accord with Seibert and Vis [43]; therefore, these records were not included in the subsequent analysis.

In total, 32 models were implemented as a result of the combination of eight watersheds, a hydrological model and four groundwater storage response structures. Twenty-five thousand simulations of each of the models were run, with a random parameter set for each watershed.

For the calibration, the logarithmic Nash-Sutcliffe efficiency (Equation (2)) was used. We used the logarithmic transformed of the Nash-Sutcliffe efficiency index ($LOG_{NSE}$) since it has been used in several studies to evaluate the performance in the low flow simulation [44–50]. The logarithmic

transformation is similar to the Box-Cox transformation used in the transformed root mean squared error (TRMSE) (see Kollat et al. [35]; van Werkhoven et al. [51]), but these transformations penalize the errors of high flows, placing increasing emphasis on low flows [52]. Therefore, the calibration was restricted to the lower part of the hydrograph [11].

$$\text{LOG}_{\text{NSE}} = 1 - \frac{\sum_{i=1}^{n}\big(\text{Ln}(Q_{\text{sim}} + \varepsilon) - \ln(Q_o + \varepsilon)\big)^2}{\sum_{i=1}^{n}\big(\ln(Q_o + \varepsilon) - \ln(\overline{Q_o} + \varepsilon)\big)^2} \tag{2}$$

where $Q_{\text{sim}}$ are the simulated streamflows, $Q_o$ are the observed streamflows, $\overline{Q_o}$ is the mean of the observed streamflows and $\varepsilon$ is a small value to avoid problems caused by observed and simulated streamflows equal to 0 [47]. Hoffmann et al. [45] suggest that $\varepsilon$ must be chosen arbitrarily as a small fraction of the mean interannual discharge (e.g., $\overline{Q_o}/40$). $\text{LOG}_{\text{NSE}}$ varies between $-\infty$ and 1, with a value equal to 1 indicating a perfect fit and values less than 1 indicating that there are differences between the simulated ($Q_{\text{sim}}$) and observed streamflows ($Q_o$).

To ascertain the sensitivity of the models to the parameters, the SAFE tool was used, separating behavioral and non-behavioral models. Models were considered behavioral above an acceptance threshold (of the performance measure), while models under the acceptance threshold were considered non-behavioral [53].

The sensitivity index was calculated from the maximum vertical distance (MVD) between the cumulative distribution functions (CDF) of the behavioral and non-behavioral models (Equation (3)) for each parameter. In accordance with Porretta-Brandyk et al. [47], $\text{LOG}_{\text{NSE}}$ values over 0.5 were considered "good" simulations; therefore, this value was set as the threshold value. Models with $\text{LOG}_{\text{NSE}}$ greater than (or equal to) 0.5 were considered behavioral, while models with values below 0.5 were considered non-behavioral.

$$\text{MVD} = \max\big|(S_{\text{bi}} - S_{\text{nbi}})\big| \tag{3}$$

where $S_{\text{bi}}$ and $S_{\text{nbi}}$ are the cumulative distribution functions for behavioral and non-behavioral models, respectively.

The MVD index varied between 0 and 1. MVD values near 1 indicate a divergence between the CDFs of the behavioral and non-behavioral models, meaning that the model was more sensitive to the parameter. By contrast, low MVD values (MVD ~0) indicate insensitivity to the parameter, since the cumulative distribution functions of the behavioral and non-behavioral models present similar shapes. Greater sensitivity suggests that processes have greater importance or influence on the results [38]. Finally, the optimal calibration parameters were obtained from the 50th percentile of the CDF of the behavioral models after the 25,000 simulations.

## 3. Results and Discussion

Table 3 presents calibration and validation $\text{LOG}_{\text{NSE}}$ values for all the implemented models. Figure 4 shows boxplots of the simulated and observed streamflows for low flows (streamflows between the 70th and 99th percentile of the duration curve) and Figure 5 shows the MVD values for the parameters associated with the response structure of each model.

**Table 3.** Model performance in the calibration and validation periods. The blue bars represent behavioral models (LOG$_{NSE}$ > 0.5) and the red bars non-behavioral models.

| | | Models | | | |
|---|---|---|---|---|---|
| | | **M1** | **M2** | **M3** | **M4** |
| CHE | Calibration | 0.81 | -0.61 | -0.82 | 0.11 |
| | Validation | 0.82 | -0.02 | -0.29 | 0.49 |
| DSL | Calibration | 0.91 | 0.17 | 0.10 | 0.62 |
| | Validation | 0.92 | 0.47 | 0.42 | 0.69 |
| CR | Calibration | 0.81 | -0.42 | -0.70 | 0.14 |
| | Validation | 0.72 | -0.17 | -0.49 | 0.22 |
| ALL | Calibration | 0.79 | -0.61 | -0.34 | -0.08 |
| | Validation | 0.59 | -0.03 | -0.09 | 0.18 |
| QL | Calibration | 0.95 | 0.85 | 0.77 | 0.89 |
| | Validation | 0.95 | 0.87 | 0.81 | 0.90 |
| CHC | Calibration | 0.93 | 0.84 | 0.80 | 0.87 |
| | Validation | 0.96 | 0.91 | 0.88 | 0.96 |
| DL | Calibration | 0.90 | 0.77 | 0.72 | 0.82 |
| | Validation | 0.92 | 0.86 | 0.80 | 0.88 |
| CA | Calibration | 0.87 | 0.77 | 0.72 | 0.83 |
| | Validation | 0.32 | 0.23 | 0.16 | 0.15 |

*3.1. Groundwater Storage Structures Performance*

In accordance with the results, differences in model performance were observed when simulating minimum flows in mountain watersheds with fractured geology and steep topography with respect to watersheds with sedimentary influence and flat topography. Model M1 presented the best calibration performance in all the watersheds independent of geological characteristics (LOG$_{NSE}$ > 0.79). Meanwhile, model M2 presented good performance (LOG$_{NSE}$ > 0.77) only in watersheds with sedimentary influence and topography with gentle slopes (e.g., CHC, DL, QL, CA), while in watersheds with volcanic influence and steep topography (e.g., CHE, DSL, CR, ALL) deficient performance was observed (LOG$_{NSE}$ < 0.5). Model M3, which had two reservoirs (one with a quick response and one slow) presented a performance similar to M2 in volcanic watersheds, with similar LOG$_{NSE}$ values (LOG$_{NSE}$ < 0.5). The deficient performance of M3 in volcanic watersheds and rugged topography can be attributed to the fact that quick response predominated in the storage-release structures of these watersheds; therefore, M3 was unable to correctly represent the hydrological processes that occurred in the mountain block system [54]. Similarly, the M4 structure did not exhibit good performance in volcanic watersheds, except in DSL, with a LOG$_{NSE}$ value of ~0.62. Its better performance in DSL could be associated with the volcano-sedimentary influence in the watershed [27], which could influence the runoff generation responses that a model with a nonlinear response is able to identify better than a model with a linear storage-release response.

In general, model performance in the validation stage was similar to that in the calibration stage (Table 3). Although CA presented a deficient performance in the validation stage. Table 4 shows the statistics mean annual corrected precipitation (through A factor), mean annual streamflow, minimum flow, 50th and 70th percentiles of the duration curve of the studied basins during the calibration and validation periods. It was observed that there was a decrease in mean annual precipitation in the validation period in all watersheds; however, the mean annual streamflow in CA increased by ~13% in the same period with respect to the calibration period, which did not occur in the other watersheds. In addition, the 70th percentile of the duration curve of CA increased in the validation period (~18%),

which would indicate that in CA there was a significant increase in minimum flows in this period that was not repeated in the other studied basins. This suggests associated problems, whether with the assessment of precipitation or the streamflow records in CA. As a consequence, the deficient performance in CA in the validation period could be related to an error in the observed data series.

**Table 4.** Streamflow and corrected precipitation (corrected precipitation by a factor) statistics for the calibration and validation periods in the studied watersheds.

| Watershed | Period | MAP (mm) | MAD ($m^3/s$) | $Q_{50}$ ($m^3/s$) | $Q_{70}$ ($m^3/s$) |
|---|---|---|---|---|---|
| CHE | Calibration | 2950 | 16.3 | 10.5 | 6.6 |
|  | Validation | 2478 | 13.2 | 7.2 | 5.7 |
| DSL | Calibration | 3091 | 18.3 | 10.1 | 5.6 |
|  | Validation | 2534 | 16.2 | 9.0 | 4.1 |
| CR | Calibration | 2417 | 95.7 | 79.4 | 50.6 |
|  | Validation | 2029 | 85.5 | 66.3 | 40.0 |
| ALL | Calibration | 2901 | 139.9 | 114.0 | 74.1 |
|  | Validation | 2774 | 127.9 | 105.0 | 74.8 |
| QL | Calibration | 2652 | 13.3 | 6.1 | 2.0 |
|  | Validation | 2314 | 12.2 | 5.7 | 2.1 |
| CHC | Calibration | 1681 | 26.3 | 8.7 | 2.8 |
|  | Validation | 1379 | 22.2 | 8.3 | 2.5 |
| DL | Calibration | 2320 | 57.0 | 22.5 | 7.7 |
|  | Validation | 1891 | 46.0 | 14.5 | 5.1 |
| CA | Calibration | 1966 | 270.0 | 167.0 | 81.1 |
|  | Validation | 1822 | 305.0 | 163.0 | 95.5 |

$Q_{70}$: 70th percentiles of the duration curve; $Q_{50}$: 50th percentiles of the duration curve.

### 3.2. Sensitivity of the Parameters Associated to Runoff Response Sub-Models

Figure 4 presents the MVD calculated for the parameters associated with each runoff response sub-model (structure) of the behavioral models ($LOG_{NSE} > 0.5$). In general, for volcanic watersheds and steep topography, behavioral models with M2, M3 and M4 were not observed. This indicates that these structures do not suitably represent or simulate the hydrological processes of groundwater storage and release in watersheds with such characteristics. Therefore, in Figure 4 only the results associated with behavioral models are shown. In addition, it is observed in the figure that the MVD values of the watersheds with volcanic influence and steep topography are higher than those of watersheds with sedimentary influence and flat topography (Figure 4). This indicates the importance of correctly representing the processes related to groundwater storage-release in watersheds with volcanic influence and topography with steep slopes.

In model M1, the most sensitive parameter in volcanic watersheds is $k_p$, which connects/controls the flow between slow and quick reservoirs. A similar result was obtained for M3, in which the parameter that distributes water between quick and slow reservoirs ($\alpha$) has high sensitivity (MVD~0.9). The greater model sensitivity to parameters could be a result of the combined effect of the fractured rock characteristics [8,27,28] and steep topography (slopes mostly greater than 50°) that these watersheds present (Figure 1b,c). Fractures can act as paths or routes [55] that allow groundwater infiltration, storage and release [22] through quick or slow runoff generation processes. Similar results were found by Rusli et al. [56], who analyzed parameter sensitivity in the Jiangwan basin in China, which has fractured geological characteristics, including cracks and faults, like those of the watersheds in this study.

In general, the parameters related to baseflow generation ($k_1$ and $k_2$ of M1, $k_1$ of M2 or $k_{01}$ and $k_{02}$ of M3) presented greater sensitivity (MVD > 0.5) than the direct runoff parameter ($k_0$). This greater MVD (sensitivity) was a result of these parameters being directly related to low-flow generation. Abebe et al. [57] mention that the greater sensitivity of the $k_2$ and $k_p$ parameters in M1 is due to the relationship between slow groundwater release processes and percolation.

In general, parameter sensitivity in watersheds with sedimentary influence and relief with low slopes (CHC, DL, QL, CA) was lower than in watersheds with volcanic influence and rugged topography (see Figure 4). In addition, unlike in mountain watersheds, in watersheds monitored in the Central Valley, wide variation in the MVD index of parameters associated with groundwater storage-release was not observed. This suggests that runoff generation processes did not predominate in these watersheds (CHC, DL, QL, CA). Therefore, a model with only one storage reservoir and less parameterization (such as M2) can suitably represent and simulate groundwater behavior in this type of watershed, with good results obtained in the simulation of low flows (as shown in Table 3). This is consistent with the results found by Fenicia et al. [11] in a study of watersheds with geological characteristics of a sedimentary origin. The sensitivity analysis also confirmed that QL was a watershed with a sedimentary influence, as the results are similar to those obtained in sedimentary watersheds (low MVD value). The observed behavior in watersheds with a volcanic influence was due to the combined influence of the geological and topographic characteristics in the groundwater storage and release process. In that sense, Sayama et al. [58] studied 17 river basins nested along the Elk River in northern California (United States); the authors mention that geology and topography largely explain the dynamic changes in groundwater storage, which is consistent with the results obtained in the present analysis.

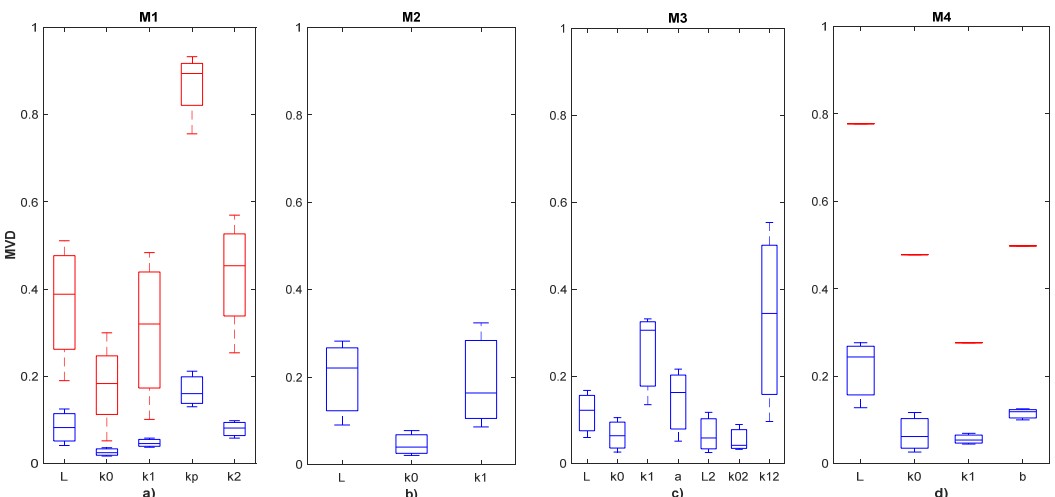

**Figure 4.** MVD variation in watersheds with (i) volcanic influence in red boxplots (CHE, DSL, CR, ALL); (ii) sedimentary influence in blue boxplots (CHC, DL, QL, CA). With models M2 and M3, no behavioral models were obtained in volcanic watersheds; therefore, no sensitivity analysis was carried out.

Figure 5 shows the comparison of streamflows between the 70th and 99th percentiles of the duration curve, the range associated with low flows [59]. It is observed in the figure that in watersheds with volcanic formation influenced by steep topographic relief (CHE, DSL, CR, ALL), the median of the streamflows simulated by M1 presented a better approximation of the median of the observed streamflows compared to the other models (see Figure 5a–d). These results are in line with those of the sensitivity analysis, given that M1 presented the greatest sensitivity to the parameters associated with slow runoff. Meanwhile, in watersheds with sedimentary influence and flat topographic relief (e.g., CHC, DL, QL) a similar distribution between observed and simulated $Q_{70}$ streamflows was seen. Models M1 and M2 presented a median near the observed values in watersheds with greater sedimentary influence and topography with gentle slopes.

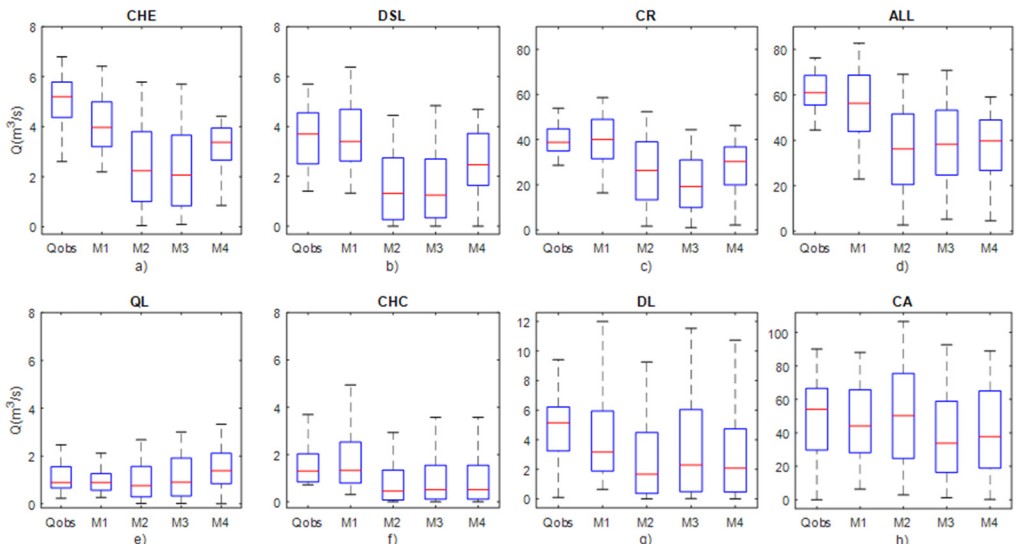

**Figure 5.** Boxplots of observed and simulated low flows ($Q_{70}$).

Complementarily, Figure 6 is presented so that the behavior of the different models in high-flow periods of two basins with different geological formations (sedimentary (a) and volcanic (b)) can be observed. Additionally, each figure shows the value of the Nash-Sutcliffe efficiency index (NSE) as a complementary measure, which is suitable for characterizing the general behavior of a hydrological model, with an emphasis on high flows. In general, all models adequately simulated the low flows of the observed streamflows in CHC (sedimentary watershed), while in DSL (volcanic watershed) only M1 adequately simulated the low flows of the observed streamflows, confirming the results obtained in the sensitivity analysis. However, according to the NSE values, all models presented a good performance in CHC (NSE > 0.74) and three models presented good behavior in DSL (NSE > 0.52). In the CHC watershed, only M2 and M4 presented good performance in the simulation of low and high flows. M1 simulated only low flows well, as it was observed that it overestimated the high flows. Similarly, M3 only showed good performance with high flows, as it overestimated the low flows. In DSL, only M1 performed well at simulating low and high flows, as M2, M3 and M4 performed poorly with low flows but well with high flows.

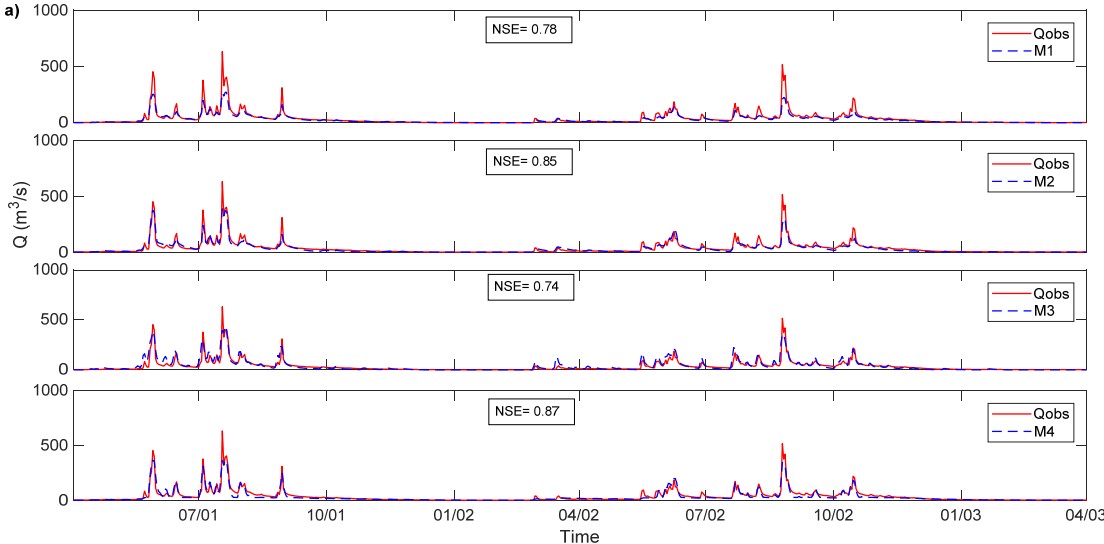

**Figure 6.** *Cont.*

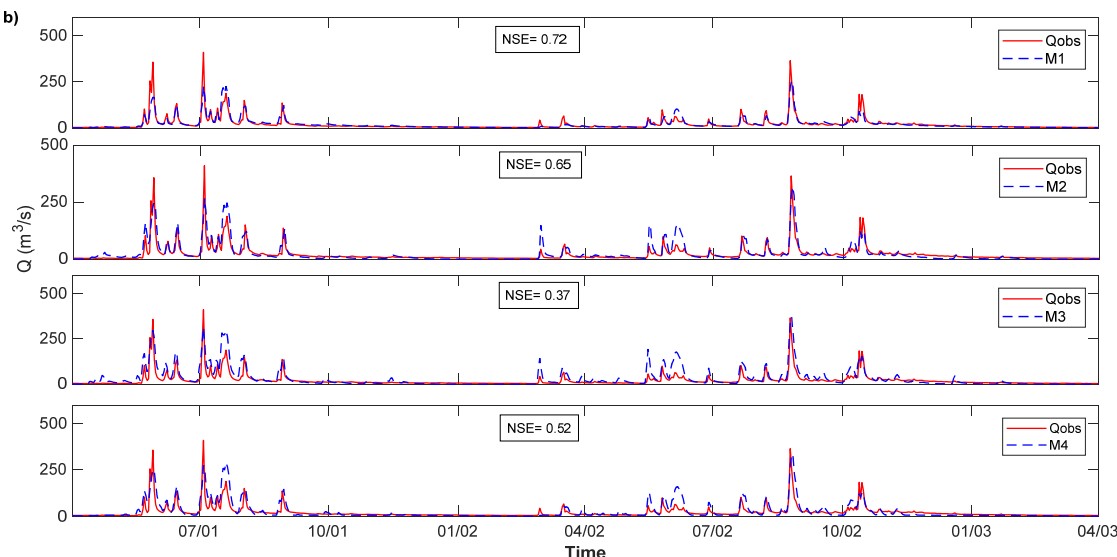

**Figure 6.** Comparison of observed and simulated streamflows for (**a**) CHC (sedimentary and flat watershed) and (**b**) in DSL (volcanic and steep watershed) (period April 2001–March 2003).

### 3.3. Influence of the Hydrological Characteristics

Regarding the hydro-meteorological characteristics of the watersheds, CHE and DSL presented similar temperature precipitation, evapotranspiration and discharge patterns (see Table 1). In addition, both watersheds had similar geology and topography. The basins to the south, CR and ALL, also presented similar hydro-meteorological, geological and topographic patterns. Although CHE and DSL presented lower precipitation and annual average flow values than CR and ALL, the models performed similarly in all these basins, which would be related to the geological similarity (percentage of volcanic formations greater than 55%) and topography (average slopes greater than 10°) in the basins. Similarly, CHC and DL presented similar hydro-meteorological, geological and topographic patterns (see Table 1). Further to the south, CA exhibited meteorological patterns similar to QL, but dissimilar average annual discharge (261 m³/s against 13.1 m³/s), which was related to differences in size. In addition, CA and QL showed similar topography, with average slopes of 2.4° and 6.0°, respectively. According to the above, hydro-meteorological patterns were also related to the model performance findings, with basins with similar topographical and meteorological characteristics presenting similar hydrological model patterns (performance and low-flow model fit).

### 3.4. M1 Model Analysis

The results show that independent of the geological and topographic characteristics of the watershed, the two-reservoir model with simultaneous responses (M1) presented the best performance simulating minimum flows of all the analyzed structures. This is consistent with other studies, in which similar results were obtained. For example, Moore [60], Pfannerstill et al. [20] and Stoelzle et al. [13] mention that models with double structures perform better than models with only one reservoir. The better performance of M1 compared to the other models can be related to its structure and parameterization. M1 contained three runoff responses controlled by parameters $k_0$ (surface runoff), $k_1$ (subsoil surface or subsurface) and $k_2$ (baseflow), as well as parameter $k_p$, which connected the two storage reservoirs. The three responses represented processes during and after rainfall periods [61] and were related to the three theoretical breakpoints (points A, B and C in Figure 7) of the recession curve of the hydrograph of a watershed mentioned in the literature [62,63]. Point B (Figure 7) indicated the start of recession flows; therefore, most streamflow input to runoff came from the aquifer. Thus, the second outlet ($Q_1$) of M1 represented primary or quick groundwater storage and release response generated by bed drainage (quick interflow, [57]). Finally, in long periods without rainfall, the surface

streamflow or quick interflow ceased [61]. This resulted in greater groundwater release from deep storage (represented by $k_2$, the third breakpoint in Figure 7). Hence, M1 represented a structure with greater flexibility to reproduce streamflow generation processes compared to other models (considering different watershed types) without under- or overparameterization that may produce an unsuitable representation of processes.

In general, in watersheds with volcanic geology and steep topography parameters, $k_1$ and $k_2$ in M1 (associated with quick and slow runoff responses) took on greater importance (sensitivity). In contrast, in watersheds of the Central Valley (sedimentary, relief with gentle slopes), parameters associated with slow runoff took on greater importance (sensitivity) (e.g., parameter $k_2$ in M2). This suggests that in watersheds characterized by both volcanic geology and rugged relief, greater emphasis on the suitable representation of streamflow generation processes is needed.

Analysis of the results of each model (Table 3) revealed that better performance was obtained in watersheds in the Central Valley than in watersheds in the Andes Mountains. These simulation results are consistent with the sensitivity analysis, as the better performance can be explained by the processes that predominate in the different studied watersheds. Due to the quick and slow streamflow generation processes (or release of water from the aquifer) in mountain watersheds, greater streamflow variability was possible. Meanwhile, the greater groundwater residence time in the watersheds of the Central Valley can generate less streamflow variability, resulting in a better simulation.

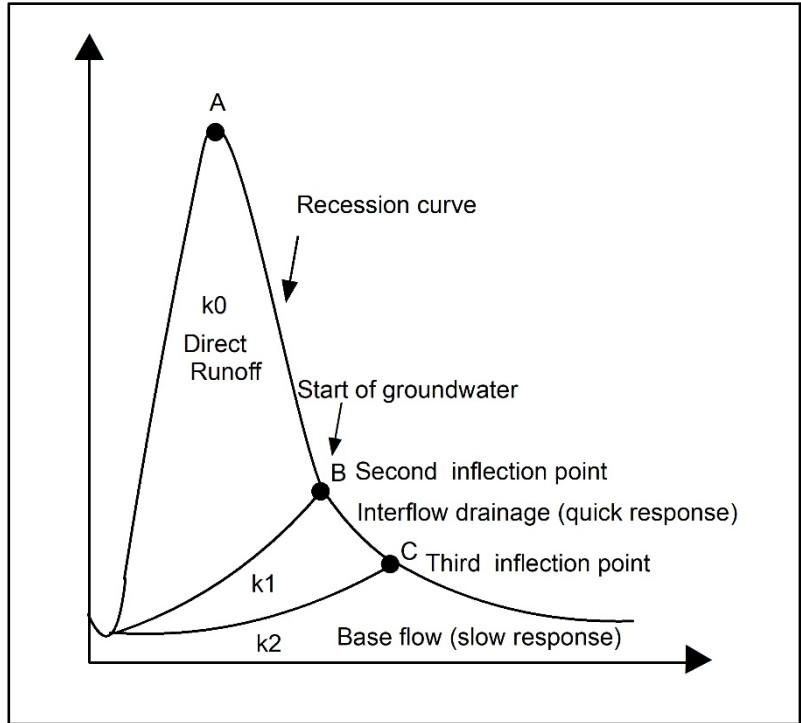

**Figure 7.** Conceptual diagram of the flow generation response of the M1 model.

## 4. Conclusions

The present study focused on analyzing different groundwater storage-release structures in watersheds with varied geological and topographic influences. It was found that, independent of these characteristics, a double-storage (two-reservoir) structure was the most suitable for simulating low flows, as it can represent and adapt to various processes that contribute to low-flow generation. Nonetheless, in areas with greater sedimentary influence and less rugged topography, there was not a predominance of quick or slow processes in low-flow generation; therefore, a simple structure with only one groundwater storage-release reservoir can (also) suitably simulate low flows in watersheds with these characteristics. In general, this study shows an analysis methodology to determine which

conceptual structure is appropriate to simulate conditions of low flows, when working in watersheds with varied geological and topographic influences.

Conceptual hydrological models require hydro-meteorological information; however, due to the availability and quality of the data, along with the combination of different characteristics in a watershed, there can be uncertainty regarding the adequate representation of the simulated processes and model choice. Increasing the number of watersheds with different characteristics can reduce the uncertainty associated with choosing an appropriate model.

Finally, it bears mentioning that the prediction of minimum flows is fundamental for different uses (e.g., industry, human consumption, hydroelectricity). Therefore, it is necessary to use groundwater models that can suitably represent the watershed characteristics involved in streamflow generation, especially in mountain watersheds that for the most part present a combination of fractured geology and rugged topography.

**Author Contributions:** V.M., J.L.A., and E.M. designed the research and analyses, V.M., J.L.A., and E.M analyzed data and performed the analyses, and V.M., J.L.A., and E.M. designed the paper, developed the discussion, and wrote the paper.

**Funding:** This research was funded by the CRHIAM, Conicyt/Fondap/15130015 project.

**Acknowledgments:** The authors thank the Dirección General de Aguas (National Water Directorate) for providing all the data for the development of this study.

**Conflicts of Interest:** The authors declare no conflict of interest.

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
