# Peer review of "Identifying a Suitable Model for Low-Flow Simulation in Watersheds of South-Central Chile: A Study Based on a Sensitivity Analysis"

_water, doi:10.3390/w11071506_

Round 1

Reviewer 1 Report

Peer review comments for the manuscript titled Identifying a suitable model for low-flow simulation in watersheds of south-central Chile: a study based on a sensitivity analysis, by Victor Parra et al.

Overview

This study tests four different configurations of the HPV conceptual hydrologic model in 8 watersheds with varying geologic and topographic characteristics in order to determine which model configuration(s) best represent low flow conditions. Of the 8 watersheds, three are volcanic and steep (CHE, DSL, and CR), one is volcanic with moderate slopes (ALL), one is volcanic and flat (QL), and three are sedimentary and flat (CHC, DL, and CA). Each watershed is represented as single hydrologic unit, and watershed areas range from 207 sq. km to 5470 sq. km.

Model inputs are temperature and precipitation. A single uncalibrated temperature threshold (TT = 0 degrees C) determines if precipitation is snow or rain, and also determines when snow melt occurs. The rate of snow melt is determined by a calibrated degree-day factor. It is not clear how potential evapotranspiration (PET) is calculated; the manuscript refers to a method that involves a correction factor, daily temperature, long-term mean PET, and monthly average temperature.

Comments

1.       Please provide additional descriptive information for each watershed such as average watershed slopes, mean annual precipitation, mean annual temperature, the fraction of precipitation falling as snow, mean annual PET, and mean annual total discharge. Please also indicate the fraction of total discharge is considered “low flow,” by month, for each watershed. Then, please add additional discussion of results for the 8 watersheds in the context of these general hydro-meteorological features.

2.       Please explain the difference between calibration and validation logNSE for the CA watershed (Table 3 and lines 247-248).

3.       Starting on line 235, the authors state that model M2 performed well only in sedimentary and flat watersheds, but Table 3 does not support this statement. Watershed QL is 100% volcanic with gentle slope, and model M2 performed well there, too.

4.       Please expand the discussion of watersheds ALL and QL relative to the other watersheds; average slopes are not listed but it appears that ALL is volcanic with moderate slope and QL is volcanic with gentle slopes. If possible, please include two additional volcanic watersheds with moderate slope, and two additional volcanic watersheds with flat slopes to permit conclusions to be drawn about these watershed characteristics.

5.       It is not clear whether model configurations M2, M3, and M4 were developed by the authors specifically for this study, or whether they are options included in the HPV model.

6.       Please clarify the method used for PET calculation.

7.       TT should be removed from Table 2 since TT was not calibrated. Please also list the calibrated parameters selected for each of the 8 watersheds.

8.       On line 182, it appears that “validation period” should be “calibration period”

9.       Does Table 3 show the best (highest) logNSE achieved after running 2500 simulations per watershed?

Author Response

Dear reviewer, please see the attachment.

Best regards.

Authors

Reviewer 2 Report

It is a very interisting work. I think it is important to see not only the boxplots but also the real hydrographs of the stations under study. I would like to have a view of the model performance als during the high flows. From the analysis was clear that the models have better results in the low  slopes and sedimentary geology.

But still the classical version of the HBV shows a better performance. Why the other versions are not operating well? Why you have studied then the different versions (M2-M4)?

How do you connect your results with the low flows?

I mean during the summer there is high irrigation activity. You mention nothing about irrigation and overpumping during the dry period.

You have there in your region plenty reservoirs and hydroelectric dams. What about the ecological flow?

How it is affecting your simulation results.

Figure 6. Conceptual diagram of the flow generation response of the M1 model.

I think figure 6 validate the classical opinion about the hydrograph recession.

Do you have to add something here new?

Why the model doent have good performance in high heighs and slopes and in volcanic rocks?

Author Response

(The authors gave the same response as above.)

Reviewer 3 Report

In this manuscript, four groundwater storage-release structures were tested for low-flow simulation in watersheds of Chile. In general, the manuscript is well-written and I think the subject is interesting. However, I cannot recommend for publication until my major concern can be properly addressed:

1.      I believe the conclusion derived in this study can be very much case sensitive. I don’t really see a clear pattern that the given conclusion (two-reservoir with three runoff responses) will valid in other places. The current format can be very useful to local engineers. However, I don’t see much scientific novelty in it.

Author Response

(The authors gave the same response as above.)

Reviewer 4 Report

This paper presents a systematic investigation on the performance and representativeness of the storage-release process modelling for simulating low flows. I believe that the topic of this paper would be of great interests to the journal readers. And the concepts and methods discussed are much needed in the water resources community. However, some points need to be elaborated to improve the readability of this paper. Therefore, I think this paper can be accepted for publication after the authors address all my comments raised below.

Line 34: The conceptual hydrological models are too simple to representing the complex hydrological processes such as the groundwater storage-release process. What about the performance of physically-based hydrological models used to simulate low flows? Please add more discussions on the physically-based hydrological models instead of the unreliable conceptual models.   

Line 57: Please explicitly state which types of models (e.g., conceptual or physical models) are used to carry out the investigation of the influence of geology and topography on the groundwater storage-release process modelling.

Line 96: Why did you use the inverse weighted distance method for interpolation? Did you perform a comparison among various interpolation methods?

Line 205: How did you determine the acceptance threshold?

Line 362: Please add limitations of this study at the end of conclusions.

Author Response

(The authors gave the same response as above.)

Round 2

Reviewer 1 Report

The authors addressed some, but not all, of my comments in the first review, and the additional information that they provided raised serious concerns.

Major comments

Using Figure 1 in combination with the discharge, precipitation, and area data in revised Table 1, it appears that most of the eight catchments receive discharge from upstream catchments, but the manuscript describes them as though they were independent catchments. Using Figure 1 with the data in revised Table 1, the runoff depths at some gages in mm/year exceed the total annual precipitation (e.g., CR and ALL), pointing to an upstream source of water, but there is no indication in the manuscript that the upstream contributing areas are modeled in this manner. Similarly, from Table 1 and Figure 1 it appears that CR discharges to CA, CHE discharges to CHC, and DSL discharges to DL – but no mention is made of this in the manuscript. Instead, the reader is left with the impression that each catchment was modeled independently, which leads to questions about how the authors accounted for discharge from an upstream source. If in fact they simulated the entire stream network, characterization of each catchment (as volcanic or sedimentary, flat or steep) must include the upstream contributing areas and it should be explicitly stated. As it stands, the conclusions regarding model performance in sedimentary/flat catchments are not supported by the data because CHC, DL, and CA include discharge from volcanic and steep contributing areas.

In my original comments, I requested more detail regarding the method of PET calculation and the authors stated that this was provided – but the additional information provided only confuses the matter. On line 144 they state that the Thornthwaite method is used to estimate PET, but in the paragraph starting on line 181, they state that daily PET is calculated using equation 1 (not Thornthwaite) based on long term mean monthly PET and C. The source of long-term monthly PET, and the value of C, are not specified.

The mountainous catchments are most likely snow-dominated, and this is supported by the hydrograph in Figure R.1, but there is no way of knowing if the same if true of the lower-elevation catchments. It is critical to know this information when evaluating performance and drawing conclusion about the four model structures. This is why I asked the authors to provide the fraction of precipitation falling as snow in each catchment in my original comments, but they declined to do so.  

Minor comments

In Figure 1a, correct the label for catchment CHE. In Figures 1a-c explain the meaning of the yellow circles versus the red triangles, and add scales to all three plots. Please add the stream network so that the reader can easily see which catchments are headwaters, and which receive discharge from an upstream source.

Line numbers

If the authors produce an additional draft, please ensure that the line numbers referenced in the responses are accurate.

Author Response

Dear reviewer,

We would like to express our gratitude to you for their constructive criticism and collaboration, which allowed us to improve the paper. Please see the attachment.

Best regards, 

The autors

Reviewer 2 Report

The manuscript was  improved

Author Response

Dear reviewer,

We would like to express our gratitude to you for your collaboration, which allowed us to improve the paper. We improved the research design and English language.

Best regards,

Reviewer 3 Report

.

Author Response

(The authors gave the same response as above.)

Reviewer 4 Report

The authors have addressed all my comments. I think the revised manuscript can be considered for publication.

Author Response

(The authors gave the same response as above.)
